# Circular RNAs in Clear Cell Renal Cell Carcinoma: Their Microarray-Based Identification, Analytical Validation, and Potential Use in a Clinico-Genomic Model to Improve Prognostic Accuracy

**DOI:** 10.3390/cancers11101473

**Published:** 2019-09-30

**Authors:** Antonia Franz, Bernhard Ralla, Sabine Weickmann, Monika Jung, Hannah Rochow, Carsten Stephan, Andreas Erbersdobler, Ergin Kilic, Annika Fendler, Klaus Jung

**Affiliations:** 1Department of Urology, Charité—Universitätsmedizin Berlin, 10117 Berlin, Germany; antonia.franz@charite.de (A.F.); bernhard.ralla@charite.de (B.R.); sabine.weickmann@charite.de (S.W.); mchjung94@gmail.com (M.J.); hannah.rochow@charite.de (H.R.); carsten.stephan@charite.de (C.S.); 2Berlin Institute for Urologic Research, 10115 Berlin, Germany; 3Institute of Pathology, University of Rostock, 18055 Rostock, Germany; andreas.erbersdobler@med.uni-rostock.de; 4Institute of Pathology, Hospital Leverkusen, 51375 Leverkusen, Germany; e.kilic@pathologie-leverkusen.de; 5Max Delbrueck Center for Molecular Medicine in the Helmholtz Association, Cancer Research Program, 13125 Berlin, Germany; annika.fendler@mdc-berlin.de

**Keywords:** clear cell renal cell carcinoma, identification of circular RNAs, experimental validation of circular RNA, diagnostic and prognostic markers, circular RNAs in a clinico-genomic predictive model, cancer-specific survival, recurrence-free survival, overall survival

## Abstract

Circular RNAs (circRNAs) may act as novel cancer biomarkers. However, a genome-wide evaluation of circRNAs in clear cell renal cell carcinoma (ccRCC) has yet to be conducted. Therefore, the objective of this study was to identify and validate circRNAs in ccRCC tissue with a focus to evaluate their potential as prognostic biomarkers. A genome-wide identification of circRNAs in total RNA extracted from ccRCC tissue samples was performed using microarray analysis. Three relevant differentially expressed circRNAs were selected (circEGLN3, circNOX4, and circRHOBTB3), their circular nature was experimentally confirmed, and their expression—along with that of their linear counterparts—was measured in 99 malignant and 85 adjacent normal tissue samples using specifically established RT-qPCR assays. The capacity of circRNAs to discriminate between malignant and adjacent normal tissue samples and their prognostic potential (with the endpoints cancer-specific, recurrence-free, and overall survival) after surgery were estimated by C-statistics, Kaplan-Meier method, univariate and multivariate Cox regression analysis, decision curve analysis, and Akaike and Bayesian information criteria. CircEGLN3 discriminated malignant from normal tissue with 97% accuracy. We generated a prognostic for the three endpoints by multivariate Cox regression analysis that included circEGLN3, circRHOBT3 and linRHOBTB3. The predictive outcome accuracy of the clinical models based on clinicopathological factors was improved in combination with this circRNA-based signature. Bootstrapping as well as Akaike and Bayesian information criteria confirmed the statistical significance and robustness of the combined models. Limitations of this study include its retrospective nature and the lack of external validation. The study demonstrated the promising potential of circRNAs as diagnostic and particularly prognostic biomarkers in ccRCC patients.

## 1. Introduction

Partial and radical nephrectomy is considered the standard of care for patients with localized clear cell renal cell carcinoma (ccRCC) [1]. Nevertheless, approximately 25% of patients experience recurrence after surgery with poor prognostic outcome within 5 years [2]. Consequently, precise stratification of recurrence risk after nephrectomy is necessary for personalized follow-up and treatment strategies. Existing prognostic models are based on conventional parameters such as tumor stage, grade, size, and resection status [1], all of which offer limited predictive accuracy for clinical outcomes [3]. Moreover, there is a broad consensus that molecular markers have, in addition to their diagnostic potential, the capacity to improve risk assessment when combined with clinicopathological factors [1,4]. At present, there are no recommended prognostic biomarkers in routine clinical use for ccRCC patients [1], though many have been evaluated experimentally [5,6,7,8,9,10].

In this regard, circular RNAs (circRNAs) are interesting potential novel biomarkers in ccRCC. CircRNAs are single-stranded, covalently closed RNA molecules without 3’- and 5’-ends and the poly(A) tail of linear RNA (linRNA). They were first identified in 2012 and are expressed widely throughout the human genome [11]; previously, they were regarded as transcriptional debris. Meanwhile, numerous studies identified their differential expression patterns in various cancers compared to normal tissue (reviewed in [12]). These expression patterns were found to be generally connected with their diagnostic, prognostic, and predictive potentials, highlighting a possible functional relevance in disease development [13,14].

Since circRNAs are a relatively new topic of scientific interest, the results of circRNA exploration in ccRCC remain limited [15,16,17,18]. Three recent reports focused on single circRNAs, which were mainly identified by database search for circRNAs fulfilling specific characteristics [15,16,18]. Only one report used three paired malignant and non-malignant kidney samples in a microarray screening for circRNAs [17]. However, with regard to the prognostic potential of circRNAs, subsequent analyses were restricted on overall survival and did not exceed the univariate Kaplan-Meier analysis. Thus, evaluation of the true prognostic potential of circRNAs in ccRCC based on genome-wide evaluation is of particular interest to apply circRNAs as biomarkers in clinical decision-making. Therefore, this study aimed to: (I) detect genome-wide differential expression patterns of circRNAs in ccRCC tissue using microarray analysis; (II) identify and validate promising circRNA candidates; and (III) evaluate the diagnostic and prognostic potentials of three circRNAs in 99 ccRCC samples and 85 adjacent normal tissue samples. Applying a combined model of both circRNA levels and clinical features, this study demonstrates the potential of circRNAs to improve prognostic value for cancer-specific (CSS), recurrence-free (RFS), and overall survival (OS).

## 2. Results

### 2.1. Patient Characteristics and Study Design

The study included ccRCC tumor samples from 99 patients and adjacent normal renal tissue samples from 85 patients undergoing radical or partial nephrectomy between 2003 and 2016 (Table 1). Samples were obtained retrospectively, and sample size was determined by a power-adapted calculation (α = 5%, power = 80%; Appendix A). The study was performed in three phases (Figure 1): (I) the discovery phase to identify differentially expressed circRNAs using a microarray screening approach; (II) the analytical validation phase to confirm the molecular characteristics of selected circRNAs and to establish “fit-for-purpose” RT-qPCR assays; and (III) the clinical assessment to evaluate the predictive value of these novel markers when applied alone and in combination with conventional clinicopathological factors.

### 2.2. Discovery of circRNAs in ccRCC Tissue Using Microarray Analysis

#### 2.2.1. Identification of Differentially Expressed circRNAs

A total of 13,261 circRNAs out of 13,617 distinct probes on the array were detected in seven matched ccRCC samples using the ArrayStar microarray approach (Appendix A). The number of circRNAs that derive from a single host gene forming multiple circRNA isoforms can vary [19]. Our microarray data revealed that approximately 50% of the detected circRNAs originated from ~75% of the 6271 host genes that produce only one or two circRNAs. However, some host genes accounted for up to 32 different circRNAs; approx. 15% of the detected circRNAs derived from host genes (3.8% of all host genes) that generate more than five circRNAs (Figure 2A). Exonic, intronic, antisense, and intergenic genomic regions can serve as sources for circRNAs. In ccRCC, 85% of the detected circRNAs derived from exonic gene sequences (Figure 2B) corresponding to data found in other human tissues [20]. Exonic circRNAs are generally assembled by one to five exons [20,21]. Analyzing the microarray data, we found 78 up-regulated and 91 down-regulated circRNAs with a higher than two-fold change (*p* < 0.05) in malignant compared to adjacent normal tissue samples (Figure 2C). This expression pattern resulted in a clear clustering of malignant vs. adjacent normal tissue using principal component analysis (Figure 2D).

#### 2.2.2. Selection of Three circRNAs for Further Evaluation

We further evaluated the differentially expressed circRNAs according to the following criteria: fold-change >4 with *p* < 0.05 and raw intensity above 500 on the microarray. Five up-regulated and eleven down-regulated circRNAs matched these ArrayStar microarray-related criteria. The nomenclature of circRNAs has not been standardized until now. In literature, different names occur depending on the reference database [12]. ArrayStar also uses its own designations. For the mentioned five up- and eleven down-regulated circRNAs, the circRNA IDs used in the databases ArrayStar and circBase [22] are summarized in Table 2. To identify circRNAs with a putative function in ccRCC initiation/progression, we selected circRNAs from host genes with putative roles in angiogenesis and hypoxia in ccRCC and other cancers including *EGLN3*, *NOX4*, and *RHOBTB3* [23,24,25,26,27]. Thus, corresponding circRNAs of the three host genes that are named according to the database circBase hsa_circ_0101692, hsa_circ_0023984, and hsa_circ_000744 were selected for further validation (Figure 2C). For greater clarity, the terms circEGLN3, circNOX4, and circRHOBTB3 are used hereafter for these circRNAs.

### 2.3. Analytical Validation of Selected circRNAs

#### 2.3.1. Experimental Confirmation of the Circularity of Transcripts

We developed RT-qPCR assays for the three selected circRNAs and their linear counterparts on the basis of SYBRGreen I. The analytical specificity of all RT-qPCR products was verified by melting curve analysis and gel electrophoresis (with Appendix A and Appendix A). Detection of circRNAs by sequencing or microarray analysis, as in our case, needs additional experimental confirmation of the circular nature of the identified transcripts to avoid false-positive results by measurement of non-circular RNA molecules with sequences similar to the specific backsplice junction [28]. Therefore, different molecular biology-based tests are recommended to validate circRNA-specific backsplice junctions [13,21,28,29]. Figure 3 summarizes our validation results based on the characteristics of circRNAs with regard to their resistance to the RNase R [13], their lack of a poly-A-tail [30], the amplification results in complementary DNA (cDNA) and genomic DNA (gDNA) using divergent and convergent primers, and the proof of the backsplice junctions by Sanger sequencing.

#### 2.3.2. Analytical Performance of RT-qPCR Assays

In addition to the analytical specificity of the established assays, the repeatability (intra-assay variation) and reproducibility (inter-assay variation) of the measurements should be characterized as decisive indicator for the performance and robustness of quantitative tests. Data in Table 3 prove that the assays and measurements are suitable for “fit-for-purpose” RT-qPCR in first clinical studies.

### 2.4. Clinical Assessment

#### 2.4.1. Differential Expression of circRNAs in Relation to Clinicopathological Factors

In this first step of the clinical assessment phase, the expression data of the three circRNAs and their linear transcripts were measured and evaluated in all samples of the studied cohort. In Figure 4, the RT-qPCR normalized expression data of these three circRNAs and their corresponding linear transcripts (named in the following with the prefix “lin”) in normal tissue and non-metastatic and metastatic primary tumor samples are shown. While the expression differences between tumor samples and adjacent normal tissue samples were significant for all circular and linear transcripts, no significant expression differences were found in primary tumors without (M0) and with (M1) metastasis.

The expression data of all three circRNAs and their linear transcripts in the tumor samples were not associated with age, sex, TNM stage, TNM-stage grouping, Fuhrman grade, surgical margin, tumor size, or metastatic status (Spearman rank correlation, Mann-Whitney U-test or Kruskal-Wallis test; *p* > 0.10; Appendix A). Only *EGLN3* showed a significant progressive down-regulation from Fuhrman grade 1 to grade 4 for circEGLN3 (from 11.9 to 9.07, 7.76, and 0.742; *p* = 0.006, Kruskal-Wallis test with Jonckheere-Terpstra trend test) and for linEGLN3, respectively (from 6.47 to 5.25, 4.51, and 0.461; *p* = 0.004).

The expression levels of the circRNAs and their linear transcripts correlated closely with each other, showing similar correlation coefficients in both malignant and adjacent normal tissue samples (circEGLN3 and linEGLN3, r_s_ = 0.742 and 0.624; circNOX4 and linNOX4, r_s_ = 0.849 and 0.851; circRHOBTB3 and linRHOBTB3, r_s_ = 0.749 and 0.849; *p* < 0.0001 in all cases). However, the ratios of the circRNAs to their linear transcripts were significantly lower (Wilcoxon test with paired samples) in the adjacent normal tissue samples than in the tumor samples (median circEGLN3/linEGLN3 of 0.68 vs. 1.57, *p* < 0.0001; circNOX4/linNOX4 of 0.79 vs. 1.16, *p* < 0.0001; circRHOBTB3/linRHOBTB3 of 0.95 vs. 0.99, *p* = 0.022).

These ratio changes and numerous significantly different correlations for each of the circRNAs with the three linear transcripts (Appendix A) support the hypothesis of differential regulatory mechanisms in normal and cancer tissue. These data encouraged us to always include the corresponding linear transcripts in subsequent investigations.

Based on the expression data, receiver-operating characteristics curve (ROC) analysis was performed to test the discriminative ability of circRNAs and the linear transcripts in differentiating between malignant and adjacent normal ccRCC tissue (Table 4). The strong discriminative potential of both circEGLN3 and linEGLN3 with regard to sensitivity, specificity, and overall accurate classification of ~95% of tissues is remarkable.

#### 2.4.2. CircRNAs as Prognostic Markers and Elaboration of RNA Signatures

To assess the prognostic value of the new markers, we defined prediction accuracy of CSS as primary and RFS and OS as secondary endpoints. The endpoints were defined as the time from the surgery until the time of the corresponding event or the last follow-up.

Kaplan-Meier analysis was used to assess the association of the expression data of the three circRNAs and linear transcripts with the outcome endpoints. For that purpose, X-tile software [33] was applied to define optimized cutoff-points (Figure 5; Appendix A and Appendix A).

For the primary CSS endpoint (Figure 5), increased expression values of both circEGLN3 and linEGLN3 were associated with better survival rates even though both transcripts were increased in malignant tissue in comparison to normal tissue (Figure 5A,B). Both circNOX4 and linNOX4 were not correlated to CSS (Figure 5C,D). Furthermore, circRHOBTB3 and linRHOBTB3 showed differing impacts on cancer-specific survival (Figure 5E,F). While high expression levels of circRHOBTB3 were associated with improved outcome, high levels of linRHOBTB3 were associated with reduced survival rates. The results were comparable using the two secondary endpoints (Appendix A). The results for the linear transcripts were validated using the The Cancer Genome Atlas Kidney Renal Clear Cell Carcinoma (TCGA-KIRC) dataset, as this data collection does not contain circRNAs. Low expression of linEGLN3 and linNOX4 as well as high expression of linRHOBTB3 were associated with shorter overall survival of TCGA ccRCC specimens (Appendix A).

In univariate Cox regression analysis, the hazard ratios of RNAs corresponded with the results from Kaplan-Meier curves (Appendix A, Appendix A, and Figure 5). Subsequently, multivariate Cox regression analysis was performed including all circRNAs and linRNAs and a backward elimination approach (entry: *p* < 0.05, removal: *p* > 0.100) was used. Only circEGLN3, circRHOBTB3, and linRHOBTB3 remained in the reduced models for all three endpoints (Table 5).

C-statistics data for the models including all six RNA variables (“full model”) compared with those obtained after backward elimination (“reduced model”), were not different (full vs. reduced model; CSS: 0.730 ± 0.060 vs. 0.726 ± 0.056, *p* = 0.863; RFS: 0.764 ± 0.057 vs. 0.735 ± 0.059, *p* = 0.478; OS: 0.741 ± 0.048 vs. 0.738 ± 0.046, *p* = 0.897; values given as AUC ± SE of the prognostic indices calculated in Cox regression analyses). We therefore used the reduced models with circEGLN3, circRHOBTB3, and linRHOBTB3 for the three outcome endpoints in the further evaluation and termed them “RNA signatures” (Table 5).

#### 2.4.3. A circRNA-Based Predictive Clinico-Genomic Model to Improve Prognostic Accuracy

Until now, clinicopathological variables are the basis to estimate the prediction of CSS, RFS, and OS after surgery and serve for benchmarking analysis of newly established tools. TNM-stage grouping including metastatic status, Fuhrman grading, tumor size, and surgical margin comprised significant clinicopathological predictors for all three outcome endpoints in Kaplan-Meier curves and univariate Cox regression analyses (Appendix A, Appendix A). These four variables were used combined in all outcome analyses and were called the “clinical model” (Table 5).

C-statistics data of the prognostic indices calculated in Cox regression analyses for the three endpoints using the “clinical model” and “RNA signature” were then compared and the results were not significantly different (Figure 6, legend: *p* values between 0.268 and 0.837). However, the results of the decision curve analysis show that the curves of the RNA-signature are always located above the curves of the “clinical model” and indicate a better accuracy. This is consistent with the recommendation that the decision curve analysis is the most informative metrics to demonstrate an incremental prognostic benefit [34]. Furthermore, the predictive outcome accuracy of the clinical models was improved upon combination with corresponding RNA signatures as shown by C-statistics and decision curve analysis (Figure 6).

Furthermore, hazard ratios of the multivariate Cox regression analysis of the clinical model variables and those of the RNA signature confirmed that the RNA signature variables remained independent factors for the prediction of the corresponding outcomes in the full combined models, but also in the reduced models after backward elimination (Table 5). The robustness of this combined classifier was supported by the fact that backward elimination of the full model “clinical model + RNA signature” did not result in a loss of predictive accuracy (full vs. reduced model; CSS: 0.832 ± 0.054 vs. 0.821 ± 0.052 *p* = 0.449; RFS: 0.818 ± 0.053 vs. 0.816 ± 0.053, *p* = 0.740; OS: 0.776 ± 0.046 vs. 0.768 ± 0.047, *p* = 0.529; values given as AUC ± SE of the prognostic indices calculated in Cox regression analyses). Furthermore, internal bootstrapping validation confirmed the statistical significance and robustness of the combined models (Appendix A). The improvement of the clinical model by including the corresponding RNA signatures was further demonstrated using the weight approach of the Akaike and Bayesian information criteria [35]. The final model including the four clinicopathological factors and the RNAs performed better than the “clinical model” with normalized probabilities of the Akaike and Bayesian criterion for CSS with 0.886 and 0.901, for RFS with 0.759 and 0.853, and for OS with 0.991 and 0.971, respectively.

### 2.5. In-silico Analysis of circRNA-miRNA-Gene Interaction

We identified potential miRNAs binding to the three circRNA candidates circEGLN3, circNOX4, and circRHOBTB3 with an algorithm provided by the CircInteractome tool [36], which is based on the database TargetScan [37]. MiRNAs were ranked according to the TargetScan context+ score. The five top-ranked miRNAs for each circRNA (all context + scores < −0.19) were chosen. Furthermore, potential gene interactions were identified for the miRNAs using the databases miRDB and TargetScan [37,38]. As cut-off values we chose a target score >90 (miRDB) and a total context++ score < −0.5. In Figure 7, only miRNA-gene interactions listed by both miRDB and TargetScan are shown.

## 3. Discussion

The current study represents hypothesis-generating research using a discovery-driven global approach [39]. We performed a genome-wide search for differentially expressed circRNAs in ccRCC tissue samples by microarray technology, the analytical and clinical validation of the three selected circRNAs circEGLN3, circNOX4, and circRHOBTB3 by RT-qPCR measurements, and successfully validated them as prognostic biomarkers in combination with conventional clinicopathological variables.

Numerous prognostic tools for ccRCC patients based exclusively on clinicopathological factors exist that have limited predictive accuracy of clinical outcome endpoints [40]. This supports the intention to include molecular markers to improve existing models [1,4]. Reports of various tools based on “omics”-markers alone or combined with clinicopathological variables have been published with promising results [1,3,4]. However, to the best of our knowledge, this study is the first to evaluate the prognostic potential of circRNAs in ccRCC tissue samples for the three clinically relevant survival endpoints of CSS, RFS, and OS.

In the discovery phase, we used microarray technology—considered to be more efficient in detecting circRNA molecules in comparison to RNA-sequencing methods [41]. The microarray contained 13 617 distinct probes, out of which 97.5% were detected in renal tissue and the principal component analysis of circRNAs clearly clustered between normal and tumor tissue. Differential expression analysis revealed 0.59% (*n* = 78) at least twofold up-regulated and 0.69% (*n* = 91) down-regulated circRNAs. These changes correspond to results observed in other solid tumors [20].

As briefly outlined in the Introduction, there have so far been few studies on circRNAs in ccRCC [15,16,17,18], which mostly focused on single circRNAs [15,16,18]. One working group investigated, based on database and literature searches, the effects of the androgen receptor and estrogen receptor beta via circHIAT1 and circATP2B1, respectively on the progression of ccRCC [15,16]. Another study examined the possible role of circABCB10 in ccRCC etiology [18]. These three circRNAs are not part of the up-/down-regulated circRNA list in our study obtained by microarray analysis. Although these studies provided interesting insights into functional aspects of circRNAs in ccRCC, their prognostic information was limited. A fourth study by Zhou al. [17] provided a list of the top 10 up- and down-regulated circRNAs based on a microarray search of three paired ccRCC tissue samples. Four of these 20 circRNAs are identical with four of the 17 top differentially expressed circRNAs, including the circEGLN3, listed in Table 2 (circBase ID: circ_0025135, circ_00331594, circ_0029340, and circ_0101692). However, circPCNXL2, which the authors reported as the circRNA with the highest up-regulation in the examined tissue samples, was not identified in our microarray analysis. This might be due to the different amount of tissue samples used for the microarray analysis or to a differing probe set on the microarray.

Despite the interesting details regarding differential expression and the ensuing discriminative potential between adjacent normal and malignant tissue (Table 4), our focus was directed at the prognostic potential of the three selected circRNAs from the circRNA profiles we obtained. In this context, the following expression particularities of the circRNAs and their linear transcripts are noteworthy: (I) we found no statistically different expression levels of circRNAs between the primary non-metastatic and metastatic tumors (Figure 4); (II) only Fuhrman grade as one relevant clinicopathological variable was associated with RNA expression (e.g., circEGLN3 and linEGLN3); and (III) we found differing and partly inverse correlations as well as changes in the abundance between circRNAs and their linear transcripts. Although the expression of most circRNAs is obviously in-line with the expression of their host genes [42], the host gene-independent expression of circRNAs, reflected in our study exemplarily in the inverse Kaplan-Meier curves of circRHOBTB3 and linRHOBTB3 (Figure 5) is of special interest. Similar findings have recently been reported also in prostate cancer and heart diseases [43,44]. We have considered these findings in the clinical assessment process by including the expression data of the linear transcripts from RNase R untreated total RNA samples in the analysis and clinical evaluation. This fact is also considered by the recently published MiOncoCirc database that includes circRNAs identified by a special capture exome RNA-sequencing protocol [45]. Using this sequencing protocol without RNase R pretreatment [46], the ratio between circular and the linear transcripts is preserved—comparable with the tissue sample—and allows a definite downstream analysis as mRNAs are not removed [45].

Thus, all these expression particularities and further the characteristics of uncorrelated differential expression in relation to conventional clinicopathological factors are specific for orthogonal biomarkers [47]. Consequently, new information can arise from the application of such biomarkers in clinical practice [48]. Indeed, multivariate Cox regression analysis showed that the RNA signatures developed with circEGLN3, circRHOBTB3, and linRHOBTB3 proved their hypothesized prognostic value for all three clinical endpoints (Table 5). Furthermore, in combination with the conventional risk factors of the clinical model, all three RNAs generally remained independent factors, also in the backward models (Table 5). Moreover, C-statistics and decision curve analysis data (Figure 6) as well as the Akaike and Bayesian information criteria and the internal bootstrap validation support the improved predictive accuracy of the combined model. We used bootstrapping for the internal validation as it is recommended as efficient method for internal validation of predictive models in favor of split-sample validation and cross-validation [49]. As already mentioned in the Results, we laid particular emphasis on the evaluation of the models using decision curve analysis as recently recommended standard to validate diagnostic/prognostic benefit [34]. Thus, the data allow us to consider circRNAs as potential prognostic biomarkers in the future to improve risk stratification of ccRCC patients after nephrectomy.

In addition to their role as promising new biomarkers in cancer and other diseases, circRNAs are also currently evaluated regarding their functional role in cancer initiation and progression (reviewed in [50]). In this context, miRNA sponging and consequential impact on the expression of cancer key genes is one of the most relevant features of circRNAs [14,51]. For example, elevated expression of circTP63 in lung squamous cell carcinoma was shown to be associated with accelerated tumor progression by sponging miR-873-3p and thus increased levels of FOXM1 [52]. In contrast, over-expression of circRNAs can also be protective against tumor progression as shown by the example of circSLCA1, which promotes the tumor suppressor PTEN via miR-130b/miR-494 sponging in bladder cancer [53]. With an in-silico analysis of the circRNA-miRNA-gene interactions of circEGLN3, circNOX4 and circRHOBTB3, we identified potential target genes of the circRNAs that might influence ccRCC development and progression (Figure 7). Interestingly, the in-silico analysis for circEGLN3 predicted a potential binding of miR-31-5p and miR-1205 to circEGLN3 and the corresponding linear RNA of EGLN3. EGLN3 is known to contribute to ccRCC initiation [25,26,54] by targeting HIF. The up-regulation of both circEGLN3 and linRNA of EGLN3 in ccRCC and the predicted interaction via miR-31-5p and miR-1205 could reflect a self-sustaining mechanism of the oncogene EGLN3. This hypothesis is supported by new reports that miRNA-31-5p acts as tumor suppressor and is down-regulated in renal cell carcinoma [55].

Furthermore, an interesting interaction was revealed for the down-regulated circRHOBTB3, which could be able to down-regulate the tumor-suppressor PTEN via absence of miR-494 sponging. The oncogenic effect of miR-494 up-regulation and subsequent PTEN inactivation is reported for many different cancers [53,56,57]. In our study, down-regulation of circRHOBTB3 is associated with poor survival outcome (Figure 6). This is especially interesting since the inverse survival outcome of circRHOBTB3 and linRHOBTB3 expression suggests a functional independency.

Nevertheless, the functional relevance of the interactions we identified based on in-silico data (Figure 7) remains to be experimentally confirmed, but this was beyond the scope of this hypothesis-generating study.

Despite the abovementioned statistical measures for bias-free analyses including the internal bootstrap-based validation and the confirmation by the Akaike and Bayesian information criteria, there are inherent limitations of this study. These include the retrospective nature of the study involving a limited number of patients selected solely based on available tissue samples, the focus on the application of biomarkers without exploring their possible molecular mechanisms, and the lack of external validation.

## 4. Materials and Methods

### 4.1. Patients and Tissue Samples

The study was approved by the local Ethics committee of the University Hospital Charité (Charité - Universitätsmedizin Berlin: EA1/134/12; approval date: 22nd June 2012) and an informed consent was obtained from patients. The study was carried out in accordance with the Declaration of Helsinki and considered the MIQE, REMARK, and STARD guidelines [58,59,60].

The study included ccRCC tumor samples from 99 patients and adjacent normal renal tissue samples from 85 patients undergoing radical or partial nephrectomy between 2003 and 2016 (Table 1). The patients were selected according to the availability of cryo-preserved tissue samples and the completeness of follow-up databased on the sample size calculation with α = 5% and a power of 80% (Appendix A). Tumors were classified by two experienced uro-pathologists (A.E., E.K.) according to the 2010 TNM classification system and the Fuhrman grading system. In total, 17 patients exhibited metastases at the time of diagnosis and 22 developed metastases within the follow-up period until November 2018 (Table 1). None of the patients received systemic therapy prior to nephrectomy. Tissue specimens were sampled immediately after nephrectomy, either snap frozen in liquid nitrogen or immersed in RNAlater solution (Qiagen, Hilden, Germany), and stored at −80 °C until analysis as described in our previous publications [61,62,63].

### 4.2. Analytical Methods

#### 4.2.1. Total RNA Samples and Their Characteristics

Total RNA was isolated from 30 to 98 mg tissue pieces. The isolation procedure from preserved tissue specimens using the miRNeasy Mini Kit (Qiagen) including an on-column DNA digestion step according to the producer's instructions, the spectrophotometric quantification (NanoDrop 1000 Spectrophotometer; NanoDrop Technologies, Wilmington, DE, USA), and the quality assessment of the total RNA samples (ratio of the absorbance at 260 nm to that at 280 nm and RNA integrity number on a Bioanalyzer 21000 (Agilent Technologies, Santa Clara, CA, USA) were detailed described in our previous publications [61,62,63]. The median ratio of 260 nm to 280 nm of the isolated RNA samples was 2.04 (95% CI, 2.03 to 2.04) and the median RNA integrity number was 7.70 (95% CI, 7.52 to 7.88). The median RNA concentration in the isolates of 30 µL nuclease-free water was 1118 (95% CI, 1029 to 1209) ng/µL). Isolated RNA samples were stored at −80 °C.

#### 4.2.2. Microarray Detection of circRNAs

Microarray analyses were performed as custom order by ArrayStar Inc. (Rockville, MD, USA) using extracted total RNA from seven paired tissue samples of non-metastasized clear cell renal cell carcinomas (ccRCC: all patients with negative lymph nodes and negative surgical margin; 1× pT1 with Fuhrman grade 2, 1× pT2 with grade 2, 3× pT3 with grade 2, 2× pT3 with grade 3). The samples were treated with RNase R to digest linear RNAs and enrich circular RNAs. The circRNAs were amplified, transcribed, and fluorescently labelled on the 3’-end using Cy3. The prepared samples were hybridized on the ArrayStar Human Circular RNA Array that is designed to detect 13.617 circRNAs. Image scanning and analysis was performed with Agilent software (Agilent scanner model G2505C, Agilent Feature Extraction software version 11.0.1.1 and Agilent GeneSpring GX). Probe intensities were normalized with quantile normalization. Differential expression analysis was carried out with R Bioconductor ‘limma’ package by computing moderated t-statistics and Benjamini–Hochberg adjusted *p* values. All the data are compiled in the accompanying separate Excel file with all additional information and annotation details (Appendix A).

#### 4.2.3. RT-qPCR Methodology and circRNA Validation Methods

RT-qPCR measurements were performed according to the recommendations of the MIQE guidelines [58]. The corresponding comments are listed in a checklist and apply for all assays (Appendix A). No template controls (NTC) and no reverse transcription controls (NRTC or no enzyme controls = NEC) were always performed and showed negative results. For cDNA synthesis, the Maxima First Strand cDNA Synthesis Kit for RT-qPCR (Thermo Fisher Scientific, Waltham, MA, USA; Cat.No. K1642) was used. All real-time qPCR runs were performed on the LightCycler 480 Instrument (Roche Molecular Diagnostics, Mannheim, Germany) in white 96-well plates (Cat.No. 04729692001) using at least technical duplicates and resulting mean values for further calculations. Maxima SYBR Green qPCR Master Mix (2X) (Thermo Fisher Scientific; Cat.No. K0252) was used. Primers were designed using the blasting tool provided by Primer3 [64] and synthesized by TIB MOLBIOL GmbH (Berlin, Germany). PPIA (peptidylprolyl isomerase A) and TBP (TATA-box binding protein) were used as normalizers [32]. Quantitative PCR data analysis was done using qbase + software, version 3.2 (Biogazelle, Zwijnaarde, Belgium). All analytical details (list of primers with their sequences, setups for all measurements, performance of RT-qPCR with melting curve analyses and agarose electrophoreses of RT-qPCR products) are compiled in Appendix A, and Appendix A. The validation methods according for the three circRNAs based on the on the RNase R approach, the cDNA synthesis with random hexamer primers and oligo(dT) primers, the Sanger sequencing, and characterization of PCR products using divergent and convergent primers [13] are summarized in Appendix A.

### 4.3. Statistics and Data Analysis

SPSS Version 25 (IBM Corp., Armonk, NY, USA) with the bootstrap module, GraphPad Prism 8.1 (GraphPad Software, La Jolla, CA, USA), and MedCalc 19.0.6 (MedCalc Software, Ostend, Belgium) were used as statistical programs. *p* < 0.05 (two-sided) was considered statistically significant. Non-parametric tests (Mann-Whitney U-test, Kruskal-Wallis test, and Spearman rank correlation) for continuous data and Chi-squared or Fisher's exact tests for categorical data were applied. To obtain optimized cutoff-points of the circRNAs and linRNAs for the outcome assessments, the software X-tile was applied [33]. Kaplan-Meier and Cox proportional hazard regression analyses were used for survival analysis. C-statistics as ROCs with AUCs decision curve analysis [65,66] served to identify the discrimination/prediction capacity of the different variables and models. The Akaike and Bayesian information criteria were used for the model evaluation [35]. GPower 3.1.9.4 [67], GraphPad StatMate 2.0 (GraphPad Software), and MedCalc were used for sample size and power determinations. For the in-silico analysis of circRNAs, the prediction tool CircInteractome [36] was used and the subsequent miRNA-gene interactions for the predicted miRNAs were analyzed with the databases miRDB [38] and TargetScan [37]. TCGA-KIRC RNAseq data were downloaded and analyzed with R (version 3.6) using the “TCGA2stat” library and the “survival” library for Kaplan-Meier analysis.

## 5. Conclusions

This study was the first to identify differentially expressed circRNA candidates using a genome-wide approach that subsequently evaluated the candidates in terms of their prognostic potential in ccRCC patients. We showed that in combination with standard clinicopathological data, circRNA-based signatures improve prognostic accuracy when predicting CSS, RFS, and OS. Furthermore, we revealed potential functional relevance of circRNAs in ccRCC by exploring circRNA-miRNA-gene interactions. CircRNAs should be considered as potential prognostic biomarkers to improve risk stratification of ccRCC patients after nephrectomy. The further exploration of circRNA functions in ccRCC might lead to new findings regarding tumor biology and pathways.

## Figures and Tables

**Figure 1 cancers-11-01473-f001:**
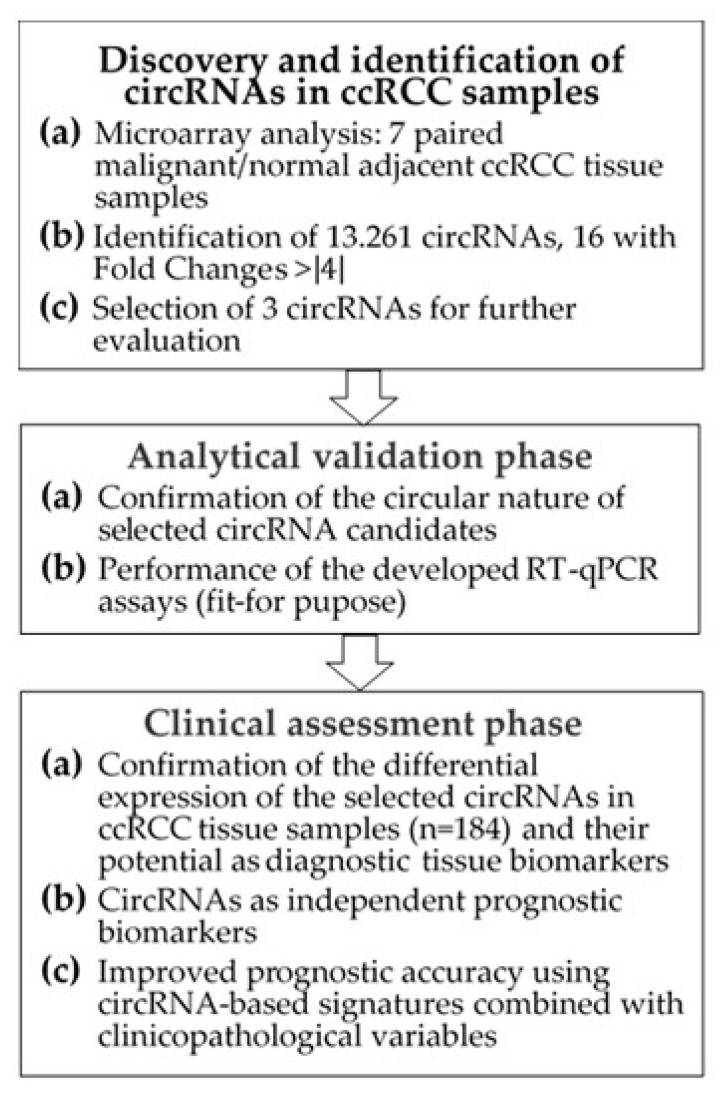
Flowchart of the study. Abbreviations: circRNA, circular RNA; ccRCC, clear cell renal cell carcinoma; RT-qPCR, reverse-transcription quantitative real-time polymerase chain reaction.

**Figure 2 cancers-11-01473-f002:**
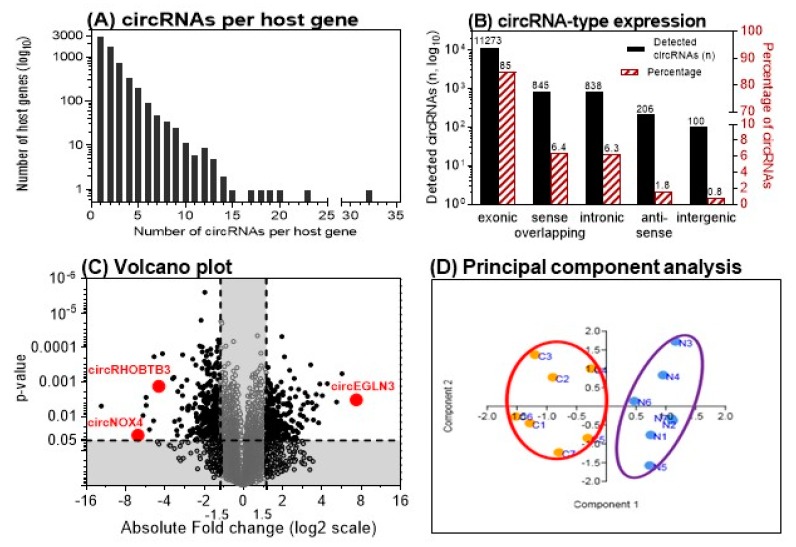
Microarray analysis results of matched clear cell renal cell carcinoma (ccRCC) tissue samples. (**A**) Number of circular RNAs (circRNAs) expressed per host gene in 7 matched ccRCC tissue samples. (**B**) Genomic origin of the detected circRNAs. (**C**) Volcano plot showing the up- and down-regulated circRNAs in malignant vs. adjacent normal tissue samples. Vertical and horizontal dashed lines indicate the thresholds of the 1.5-fold changes and the *p*-values of 0.05 in the *t*-test. The positions of the three detailed examined circRNAs in this study are marked. (**D**) Principal component analysis with the left cluster of tumor samples (C1–C7) and the right cluster with the paired adjacent normal tissue samples (N1–N7). (A and B adapted from [12]).

**Figure 3 cancers-11-01473-f003:**
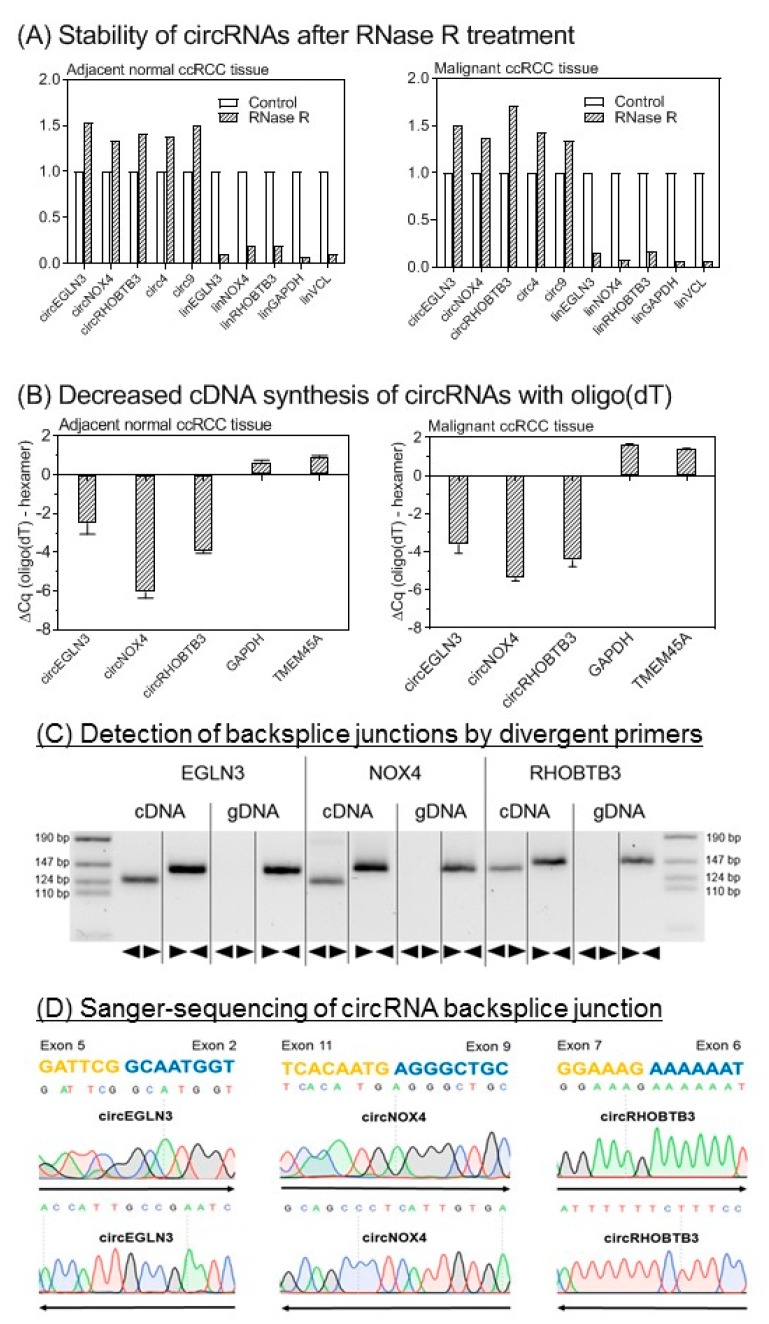
Analytical validation of the circular nature of circEGLN3, circNOX4, and circRHOBTB3. (**A**) Stability of circular RNAs (circRNAs) after RNase R treatment. CircRNAs are stable while linear mRNAs are degraded when treated with RNase R. Glyceraldehyde 3-phosphate dehydrogenase (GAPDH) mRNA (glyceraldehyde-3-phosphate dehydrogenase), VNL mRNA (vinculin), circ4, and circ9 were used as additional controls [13,31]. Data of triplicate experiments normalized to controls without RNase treatment are presented. (**B**) Random hexamer vs. oligo(dT) primers for cDNA synthesis. Random hexamer primers are used for amplification during cDNA synthesis of circRNAs as covalently closed structures of circRNAs lack a poly-A-tail. The binding capacity of oligo(dT) primers is therefore reduced without polyadenylated binding sites. In consequence, quantitation cycle (ΔCq) values in RT-qPCR are distinctly reduced when using random hexamer primers in comparison to oligo(dT) primers for circRNA cDNA synthesis. GAPDH and TMEM45A (transmembrane protein 45A) were used as mRNA controls. Significantly different mean values between the six circRNA samples and four mRNAs of GAPDH and TMEM45A as mRNA controls (mean values: −4.33 vs. 1.18, *p* < 0.0001) confirmed this characteristic feature of circRNAs. Different mean values of the circRNAs between the tissues were not observed (*p* > 0.799). (**C**) Gel electrophoresis of PCR products obtained from cDNA and genomic DNA (gDNA). Divergent (◄►) primers used for circRNA measurements amplify sequences only in cDNA. Convergent (►◄) primers show amplification of circRNA composing exons in cDNA and gDNA. (**D**) Base sequence of circRNA backsplice junction pictured by Sanger sequencing. Sanger sequencing was performed with forward (→) and reverse (←) primers. Methodical details for all here listed experiments are described in section “Material and Methods” and in Appendix A.

**Figure 4 cancers-11-01473-f004:**
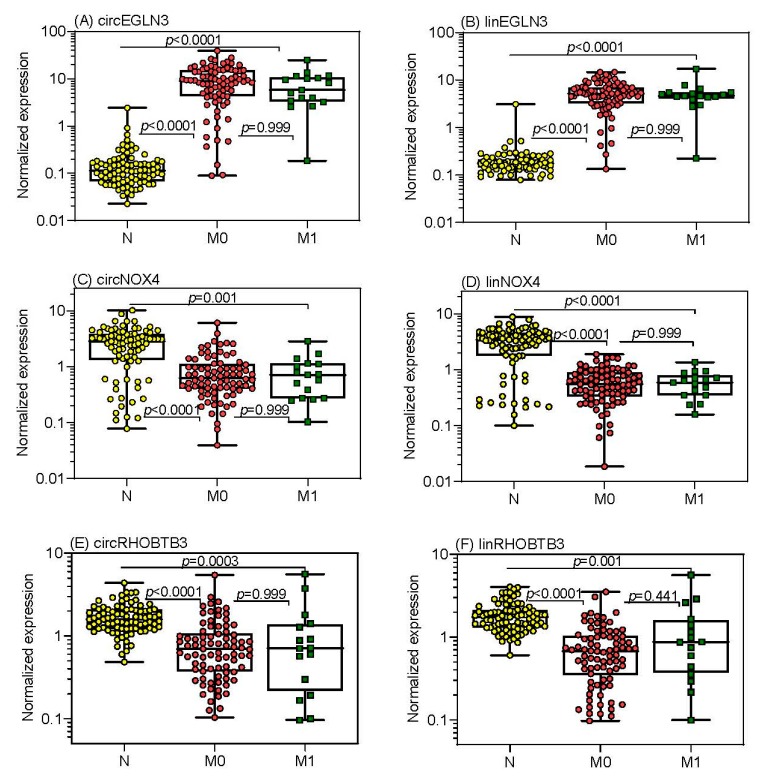
Expression of circular RNAs (circRNAs) and the linear transcripts of their host genes in tissue samples from patients suffering from clear cell renal cell carcinoma. Expression data of (**A**) circEGLN3, (**C**) circNOX4, and (**E**) circRHOBTB3 as well as the corresponding linear transcripts of the host genes (**B**) EGLN3, (**D**) NOX4, and (**F**) RHOBTB3 are shown in adjacent normal tissue distant from tumor (*n* = 85), in tissue from non-metastatic (M0, *n* = 82) and metastatic primary tumors (M1, *n* = 17) of patients with clear cell renal cell carcinoma at the time of surgery. CircRNA and linRNA expression ratios of M0 (*n* = 60) and M1 (*n* = 16) tissue samples in relation to their paired adjacent normal tissue samples did not statistically differ (*p* values between 0.302 to 0.712). PPIA (peptidylprolyl isomerase A) mRNA and TBP (TATA-box binding protein) mRNA were used as normalizers. Boxes in the box-and whisker plots represent the lower and upper quartiles with medians, whiskers illustrate the entire range of the samples. Significant differences between the study groups were estimated by the Kruskal-Wallis test with multiple comparisons corrected according to Holm-Sidak.

**Figure 5 cancers-11-01473-f005:**
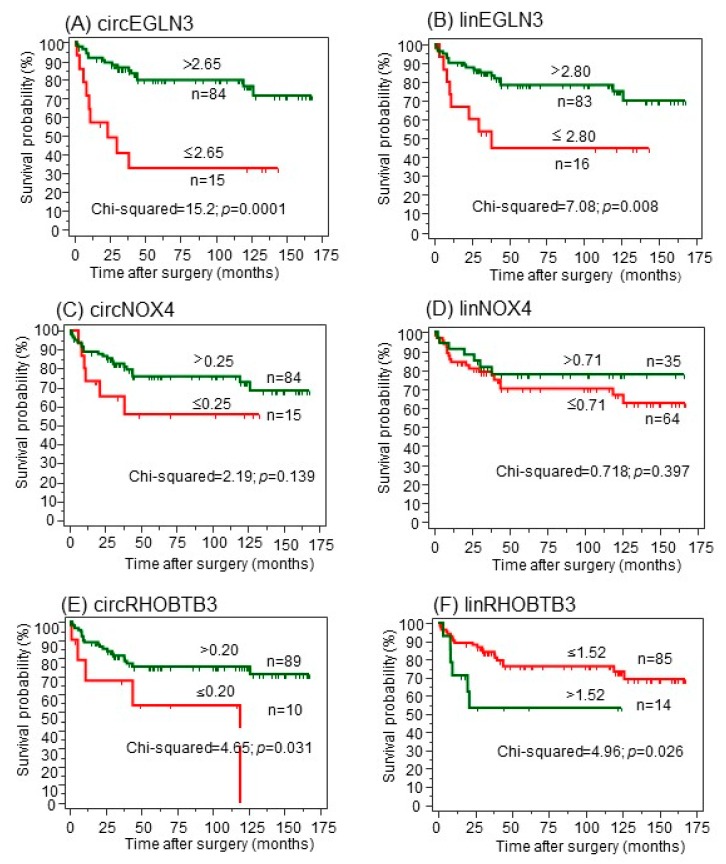
(**A**–**F**) Kaplan-Meier analysis of circular RNAs (circRNAs) and their linear transcripts with regard to cancer-specific survival after surgery. CircRNAs were dichotomized using the optimized cutoffs indicated by software X-tile [33] to discriminate between deceased and alive. Green curves represent patients with expression values above the cutoff; red curves represent patients with values equal or below the cutoff. The number of patients in the dichotomized groups and the cutoffs are indicated at the curves. The log-rank test was used to confirm significant differences between the survival probabilities.

**Figure 6 cancers-11-01473-f006:**
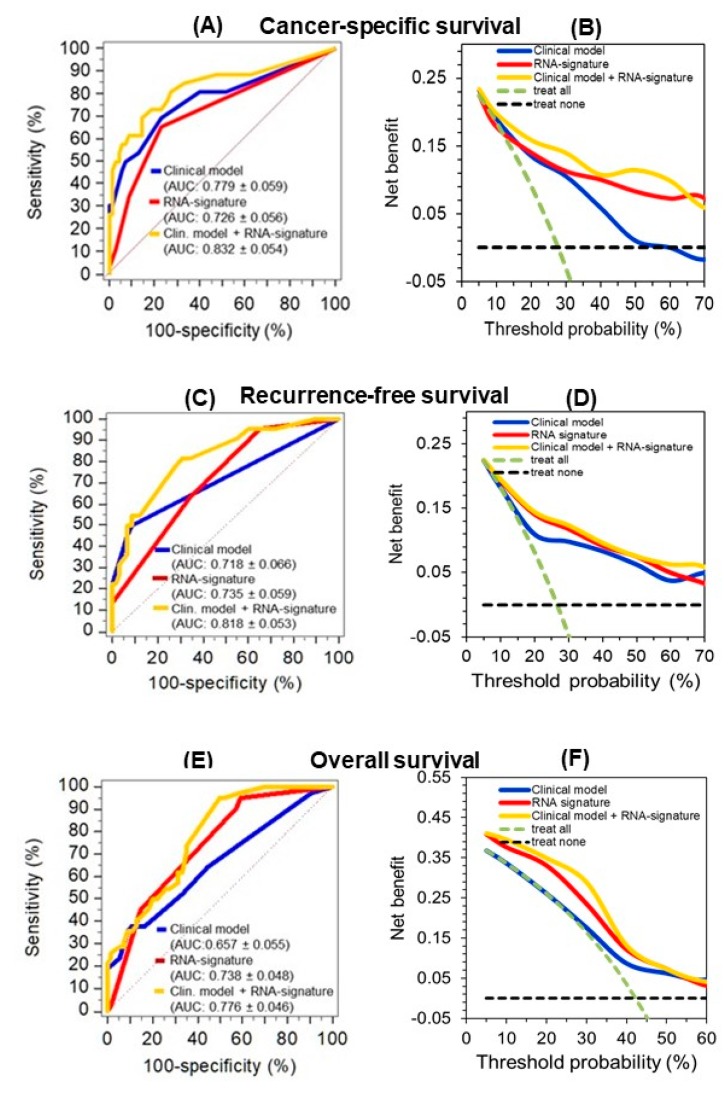
Improved predictive accuracy of cancer-specific (CSS), recurrence-free (RFS), and overall survival (OS) in a model with clinicopathological variables by including a circular RNA (circRNA) based signature. (**A**, **C**, and **E**) C-statistics curves of the prognostic indices of Cox regression analysis for the three endpoints using the models indicated in Table 5 (“clinical model”, “RNA signature”, and the combination of both (“clinical model + RNA signature”) as well as (**B**, **D**, and **F**) the corresponding curves of decision curve analysis are shown here. The AUC values were not significantly different between the clinical model and the RNA signature for CSS (*p* = 0.452), RFS (*p* = 0.837), and OS (*p* = 0.268). The improved predictive accuracy of combining the “RNA signature” with the pure “clinical model” is shown by the increased AUC values indicated in parentheses next to the models in the subfigures. Curves in the decision curve analysis confirmed the benefit of combining the RNA signatures with the “clinical model.”.

**Figure 7 cancers-11-01473-f007:**
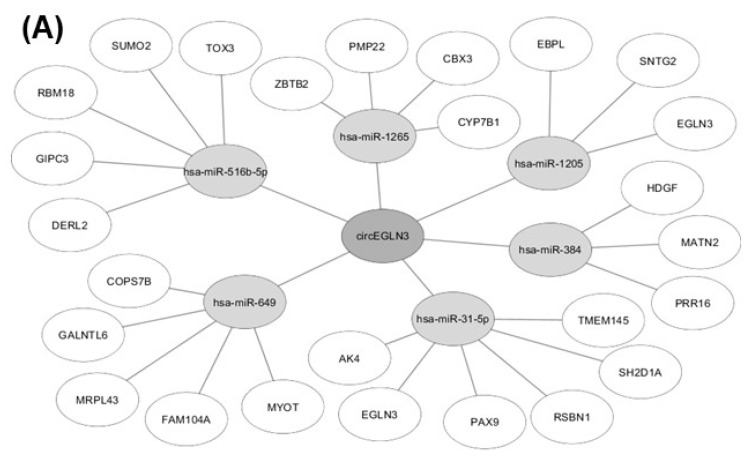
Results of in-silico analysis of circRNA-miRNA-gene interaction. MiRNAs with potential binding sites to (**A**) circEGLN3, (**B**) circNOX4, and (**C**) circRHOBTB3 were identified using CircInteractome [36] and subsequent miRNA-gene interactions were analyzed with the databases miRDB [38] and TargetScan [37].

**Table 1 cancers-11-01473-t001:** Characteristics of the ccRCC patients.

Characteristics	Total	ccRCC,	ccRCC,	*p*-value ^b^
		Non-Metastatic ^a^	Metastatic ^a^	
Patients, no.	99	82	17	
Sex, female/male; n (%)	30/69	25/57 (30/70)	5/12 (29/70)	1
Age, yrs, median (range)	65 (36–87)	65 (36–87)	65 (37–78)	0.982
Pathological stage, no. (%)				
pT1	43	40 (49)	3 (18)	0.004
pT2	8	8 (10)	0 (0)	
pT3	47	34 (41)	13 (76)	
pT4	1	0 (0)	1 (6)	
TNM stage grouping, no. (%) ^c^				
I	40	40	-	
II	8	8	-	
III	34	34	-	
IV	17	0	17	
Tumor size, median mm (range)	57 (20–220)	50 (20–220)	75 (35–150)	0.029
Surgical margin, no. (%)				
R0	85	73 (89)	12 (71)	0.138
R1/2	11	7 (9)	4 (23)	
Unclassified	3	2 (2)	1 (6)	
Fuhrman grade, no. (%)				
G1	9	8	1	0.025
G2	68	60	8	
G3	20	12	8	
G4	2	2	0	
Events during follow-up, n, (%)				
Metastasis	-	22 (27)	-	
Cancer-specific death	26 (26)	14 (17)	12 (71)	<0.0001
Overall death	42 (42)	30 (37)	12 (71)	0.014
Survival, months, median/mean (95% CI) ^d^				
Cancer-specific	125 (111–139)	140 (127–153)	27.2 (14.1–40.4)	
Recurrence-free	-	127 (113–141)	-	
Overall	126 (90.4–166)	160 (103–160)	11.8 (8.2–39.9)	

Abbreviations: ccRCC, clear cell renal cell carcinoma; G, histopathological grading according to Fuhrman; pT, pathological tumor classification; R, surgical margin classification; CI, confidence interval. ^a^ Imaging was used to assess the presence/non-presence of metastases before surgery. ^b^ Calculated with Fisher's exact test, Chi-squared test or Mann-Whitney *U* test between the two groups. ^c^ TNM stage grouping according to UICC classification system. ^d^ Survival data obtained from the Kaplan-Meier analyses using the software MedCalc. The median survival (overall survival) corresponds to the time at which the survival probability reaches 50% or below. As the cancer-specific survival and recurrence-free survival did not reach this value in the non-metastatic cohort, the mean survival time was calculated (as area under the survival curve in the total follow-up interval) for both groups for comparison purposes.

**Table 2 cancers-11-01473-t002:** List of circular RNAs (circRNAs) with at least a fourfold differential expression between the matched malignant vs. adjacent normal tissue samples (*n* = 7) in the microarray discovery study phase. The three circRNAs selected for further examination in this study are marked in bold letters.

circRNA in Manuscript	circRNA ID in ArrayStar ^a,b^	circRNA ID in circBase ^a,c^	Fold Change Expression in Tumor vs. Normal Tissue (*p*-value)	Best Transcript	Official Gene Symbol
Up-regulated circRNAs					
**circEGLN3**	**circRNA_405198**	**circ_0101692**	**7.32 (0.0033)**	**NM_022073**	**EGLN3**
-	circRNA_101202	circ_0029340	5.68 (0.0006)	NM_005505	SCARB1
-	circRNA_101341	circ_0031594	5.16 (0.0038)	NM_022073	EGLN3
-	circRNA_101803	circ_0003520	4.39 (0.0011)	NM_018092	NETO2
-	circRNA_103980	circ_0006528	4.01 (0.0024)	NM_138492	PRELID2
Down-regulated circRNAs					
-	circRNA_103093	circ_0060937	−12.3 (0.0049)	NM_000782	CYP24A1
-	circRNA_101120	circ_0027821	−6.73 (0.0342)	NR_024037	RMST
**circNOX4**	**circRNA_100933**	**circ_0023984**	**−6.44 (0.0475)**	**NM_016931**	**NOX4**
-	circRNA_100562	circ_0006577	−5.87 (0.0093)	NM_012425	RSU1
-	circRNA_031282	circ_0031282	−5.58 (0.0028)	NM_012244	SLC7A8
-	circRNA_103091	circ_0060927	−5.58 (0.0048)	NM_000782	CYP24A1
-	circRNA_023983	circ_0008350	−5.26 (0.0239)	NM_016931	NOX4
-	circRNA_1011001	circ_0025135	−4.89 (0.0132)	NM_001038	SCNN1A
-	circRNA_035435	circ_0035435	−4.84 (0.0002)	NM_032866	CGNL1
**circRHOBTB3**	**circRNA_007444**	**circ_0007444**	**−4.45 (0.0013)**	**NM_014899**	**RHOBTB3**
-	circRNA_101528	circ_0035436	−4.18 (0.0001)	NM_032866	CGNL1

^a^ The obligatory prefix hsa_ was omitted to facilitate the readability. ^b^ More detailed annotations including source, chromosome localization, strand, circRNA type, and sequences are listed for all detected circRNAs in the Appendix A. ^c^
http://www.circbase.org and [22].

**Table 3 cancers-11-01473-t003:** Repeatability and reproducibility of RT-qPCR measurements.

RNA	Repeatability ^a^		Reproducibility ^b^	
	Cq Value Mean (%RSD)	Concentration (AU) Mean (%RSD)	Cq Value Mean ± SD (%RSD)	Concentration (AU) Mean ± SD (%RSD)
circEGLN3	23.47 (0.284)	1.185 (4.51)	22.49 ± 0.138 (0.62)	2.132 ± 0.199 (9.33)
circNOX4	24.16 (0.493)	0.808 (9.35)	22.77 ± 0.108 (0.47)	1.118 ± 0.082 (7.30)
circRHOBTB3	25.64 (0.459)	0.0268 (9.27)	24.81 ± 0.190 (0.76)	0.036 ± 0.005 (14.2)
linEGLN3	20.83 (0.521)	32.72 (7.00)	23.93 ± 0.042 (0.18)	1.658 ± 0.046 (2.75)
linNOX4	27.65 (0.405)	0.204 (7.67)	25.50 ± 0.046 (0.18)	0.483 ± 0.015 (3.10)
linRHOBTB3	24.90 (0.214)	1.901 (3.43)	23.43 ± 0.147 (0.63)	3.791 ± 0.386 (10.2)
PPIA	19.33 (0.329)	32.01 (4.16)	19.18 ± 0.081 (0.42)	33.36 ± 1.745 (5.23)
TBP	25.18 (0.331)	2.330 (5.07)	24.99 ± 0.104 (0.42)	2.423 ± 0.156 (6.44)

Abbreviations: Cq, quantitation cycle; AU, arbitrary units; %RSD, percent relative standard deviation; SD, standard deviation; PPIA, peptidylprolyl isomerase A; TBP, TATA-box binding protein. PPIA and TBP served as reference genes [32]. ^a^
*n* = 21; %RSD was calculated from duplicate measurements using the root mean square method based on Cq values and calculated concentrations, respectively. ^b^
*n* = at least 8; %RSD (Cq) corresponds to the percent relative standard deviation calculated on the basis of the Cq values. %RSD (Concentration) corresponds to the percent relative standard deviation calculated on the basis of the normalized relative quantities (arbitrary units).

**Table 4 cancers-11-01473-t004:** Receiver-operating characteristic curve analyses of circRNAs and their linear transcripts to discriminate between adjacent normal and malignant tissue.

RNAs	AUC (95% CI)	*p*-Value Different to AUC = 0.5	Differentiating Ability at the Youden Index ^a^		% Overall Correct Classification
			Sensitivity (95% CI)	Specificity (95% CI)	
circEGLN3	0.98 (0.95–0.99)	<0.0001	95 (89–99)	95 (88–99)	94.6
circNOX4	0.81 (0.74–0.86)	<0.0001	91 (83–96)	71 (60–80)	80.4
circRHOBTB3	0.82 (0.76–0.87)	<0.0001	72 (62–80)	91 (82–96)	69.0
linEGLN3	0.98 (0.96–0.99)	<0.0001	96 (89–98)	99 (94–100)	95.7
linNOX4	0.85 (0.79–0.90)	<0.0001	99 (95–100)	78 (67–86)	88.0
linRHOBTB3	0.86 (081–0.91)	<0.0001	72 (62–80)	94 (87–98)	75.0
circEGLN3 +linEGLN3 ^b^	0.99 (0.96–1.00)	<0.0001	95 (89–98)	99 (94–100)	95.7

Abbreviations: AUC, area under the receiver-operating characteristics curve; CI, confidence interval. ^a^ The Youden index as a measure of overall diagnostic effectiveness is calculated by [(sensitivity + specificity) − 1]. When equal weight is given to sensitivity and specificity of a test, the cutoff at the maximum value of this index, which graphically corresponds to the maximum vertical distance between the ROC curve and the diagonal line, is referred to as optimal criterion. ^b^ Calculated by binary logistic regression.

**Table 5 cancers-11-01473-t005:** Multivariate Cox Proportional Hazard Regression Analyses of Different Prediction Models for Outcome after ccRCC Nephrectomy ^a^.

Variable ^b^	Cancer-Specific Survival		Recurrence-Free Survival		Overall Survival	
	HR (95% CI)	*p*-Value	HR (95% CI)	*p*-Value	HR (95% CI)	*p*-Value
RNA signature. ^c,d,e^						
circEGLN3	0.24 (0.10–0.57)	0.001	0.53 (0.18–1.03)	0.074	0.49 (0.23–0.95)	0.037
circRHOBTB3	0.26 (0.09–0.73)	0.010	0.14 (0.04–0.49)	0.003	0.15 (0.05–0.49)	0.002
linRHOBTB3	2.57 (0.95–6.90)	0.062	11.1 (2.79–43.8)	0.001	4.46 (1.52–13.0)	0.006
Clinical model						
Tumor stage grouping (III+IV/I+II)	3.54 (1.16–10.8)	0.027	0.79 (0.25–2.48)	0.685	2.02 (0.95–4.27)	0.064
Fuhrman grading (3+4/1+2)	2.68 (1.03–7.02)	0.044	13.4 (4.06–44.3)	<0.0001	3.00 (1.32–6.82)	0.009
Surgical margin (R1/R0)	2.82 (1.05–7.59)	0.040	7.09 (1.75–28.7)	0.006	2.26 (0.91–5.59)	0.078
Tumor size (≥7 cm<)	1.10 (0.42–2.87)	0.838	1.03 (0.35–2.97)	0.963	0.99 (0.46–2.14)	0.988
Clinical model + RNA signature						
Tumor stage grouping (III+IV/I+II)	3.16 (0.93-10.8)	0.066	0.67 (0.19–2.37)	0.536	2.81 (1.23–6.42)	0.014
Fuhrman grading (3+4/1+2)	2.04 (0.73–5.74)	0.177	9.98 (2.99–33.4)	0.0002	2.25 (0.99–5.12)	0.053
Surgical margin (R1 vs. R0)	3.97 (1.28–12.3)	0.017	3.48 (0.76–15.9)	0.109	2.12 (0.87–5.18)	0.099
Tumor size (≥7 cm<)	1.16 (0.44–3.11)	0.762	1.22 (0.36–4.13)	0.746	0.76 (0.36–1.64)	0.490
circEGLN3	0.33 (0.13–0.79)	0.014	0.69 (0.22–2.19)	0.532	0.51 (0.24–1.09)	0.084
circRHOBTB3	0.25 (0.08–0.80)	0.019	0.21 (0.05–0.94)	0.041	0.16 (0.05–0.53)	0.003
linRHOBTB3	3.84 (1.35–10.9)	0.012	7.71 (1.55–38.3)	0.013	5.26 (1.74–15.9)	0.003
Clinical model + RNA signature after backward elimination						
Tumor stage grouping (III+IV/I+II)	3.98 (1.32–12.0)	0.014	–	–	2.89 (1.39–5.99)	0.005
Fuhrman grading (3+4/1+2)	–	–	8.53 (3.08–23.6)	<0.0001	2.52 (1.15–5.53)	0.021
Surgical margin (R1 vs. R0)	5.68 (2.09–15.4)	0.0007	3.54 (1.02–12.3)	0.047	–	–
Tumor size (≥7 cm<)	–	–	–	–	–	–
circEGLN3	0.28 (0.12–0.68)	0.005	–	–	0.50 (0.24–1.05)	0.067
circRHOBTB3	0.21 (0.07–0.65)	0.007	0.18 (0.04–0.75)	0.018	0.17 (0.05–0.56)	0.004
linRHOBTB3	3.59 (1.28–10.0)	0.015	8.19 (1.81–37.1)	0.006	5.37 (1.80–16.1)	0.003

Abbreviations: ccRCC, clear cell renal cell carcinoma; CI, confidence interval; G, histopathological grading according to Fuhrman; HR, hazard ratio; R, surgical margin classification. ^a^ The multivariate analysis included the RNA signature combination of circEGLN3, circRHOBTB3, and linRHOBTB3 after univariate analysis with a backward elimination approach of the six RNA and all clinicopathological factors of univariate analysis with *p* values < 0.05 (Appendix A). ^b^ RNAs were dichotomized using the X-tile program [33] at the best threshold to discriminate between dead and alive in cancer-specific and overall survival, respectively as well as between recurrence and recurrence-free situation. These outcome-specific thresholds are indicated in the footnotes c, d, and e and correspond to those in the Kaplan-Meier curves. Clinicopathological variables are given with their categorized criteria in brackets. ^c^ Cutoffs for cancers-specific survival: circEGLN3 (2.65), circRHOBTB3 (0.20), and linRHOBTB3 (1.52). ^d^ Cutoffs for recurrence-free survival: circEGLN3 (2.10), circRHOBTB3 (0.50), and linRHOBTB3 (0.72). ^e^ Cutoffs for overall survival: circEGLN3 (2.73), circRHOBTB3 (0.27), and linRHOBTB3 (0.60).

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
