# Peer review of "Circular RNAs in Clear Cell Renal Cell Carcinoma: Their Microarray-Based Identification, Analytical Validation, and Potential Use in a Clinico-Genomic Model to Improve Prognostic Accuracy"

_cancers, 2019, doi:10.3390/cancers11101473_

Round 1

Reviewer 1 Report

This was a well crafted study, with excellent design and downstream analyses. I very much enjoyed reading this manuscript.

I have three very minor items that I think the authors can address easily (should they so wish), they don't necessarily have to be included in the main manuscript

Item 1. Page 15. Line 343 "In Figure 4..." this should be "In Figure 7...?"

Item 2. I recommend that the authors re-examine their genes by using TCGA dataset analysis. This can be done using a plethora of potential websites including: KM-Plot, ALCAN or ProgGeneV2. At least for me, ELGN3 panned out in KM-PLot (RNA-Seq analysis), so may add additional weight to your already impressive data.

Item 3. Could the authors discuss (or even test), how the observations for ELGN3 and circELGN3 in ccRCC, might be with respect to EGLN3-AS1 in these tissues?

Reviewer 2 Report

Franz et al.

In this manuscript, the authors aim at assessing whether micro-array quantification of specific circular RNAs could help diagnose/prognose post-surgery ccRCC outcomes. 78 upregulated and 91 downregulated circRNAs are identified among 13617 candidates tested. Among those, three are chosen for further validation (EGLN3, RHOBT3 and NOX4) due to their putative roles in angiogenesis and hypoxia.

Among the 5 up-regulated and 11 down-regulated, it is not mentionned whether the RNAs that have been identified in other ccRCC studies are found (references 15 to 18). This is a major week point of the study, in which those prior findings are never critically discussed.

The study contains lots of statistics for which the application conditions have been well checked. However this profusion masks somehow the real predictive value of the circ/linRNAs indicators. Figure 4 for example lacks a positive control which could show that in M0 or M1 cells no differential expression with N cells. Also the ROC analysis in figure 6 does not show the net benefit of the RNA signature alone, but only combined to the clinical model. It is very important to show and discuss the contribution of the RNA signature alone to evaluate the potential of using the markers studied in this paper.

A follow-up version of the paper should correct those aspects and allow for using a lesser amount of statistics, which blurs the message and makes the overall manuscript difficult to read by losing its main focus.

Detailed comments.

Line 57. circRNAs are not a new class of RNAs. What is new is eventually their interest in cancer prognotics. Line 59. 5'-3' polarization is never abolished in RNA or DNA, due to the intrinsic structure of individual nucleotides. The 5' and 3' termini are lost following circularization. Line 107-108. It is OK that 50% of detected circRNAs originate from genes producing only 1 or 2 circRNAs. But why do not include the "at least 5 other circRNAs" in the 15% derived from the same gene. Line 138. Mention to figure 2A seems to be fraught. Line 231-236. Sentence too long. Split to clarify meaning. Line 263-268. The Kaplan-Meier curves relative to NOX4 are not commented in the paragraph. Line 304-308. The text does not restitute the figure 6 exactly. It is not clear from the text that only the addition of the RNA signature on top of the clinical model leads to a partial improvement of the outcome prediction. So Figure 6 should display the Net benefit of the RNA model alone (in red) like on the sensitivity plots. Line 343. figure 7 is called, not figure 4. Table 2 should confront the results obtained in other studies (ref 15-18). Figure 3. Panel B lacks distinguish between normal and malignant of tissues, in opposition to the comments in the legend. No p-value provided. Figure 5. Indicate color code of curves in the legend, and on the graph the X-tile threshold. It would be appropriate to comment on panels C and D in the text. Figure 6. It would be consistant to show "RNA-signature" on panels showing Net benefit to illustrate the text fom page 13.

Round 2

Reviewer 2 Report

My comments and questions have been fully addressed.